# Molecular Targets in Salivary Gland Cancers: A Comprehensive Genomic Analysis of 118 Mucoepidermoid Carcinoma Tumors

**DOI:** 10.3390/biomedicines11020519

**Published:** 2023-02-10

**Authors:** Maroun Bou Zerdan, Prashanth Ashok Kumar, Daniel Zaccarini, Jeffrey Ross, Richard Huang, Abirami Sivapiragasam

**Affiliations:** 1Department of Internal Medicine, SUNY Upstate Medical University, Syracuse, NY 13210, USA; 2Division of Hematology/Oncology, Department of Medicine, SUNY Upstate Medical University, Syracuse, NY 13210, USA; 3Department of Pathology, SUNY Upstate Medical University, Syracuse, NY 13210, USA; 4Foundation Medicine, Inc., Morrisville, NC 27560, USA; 5Departments of Pathology and Urology, SUNY Upstate Medical University, Syracuse, NY 13210, USA

**Keywords:** salivary gland tumors, comprehensive genomic profiling, mucoepidermoid carcinoma, immunotherapy, next-generation sequencing

## Abstract

Introduction: Salivary gland carcinomas (SGC) are histologically diverse cancers and next-generation sequencing (NGS) to identify key molecular targets is an important aspect in the management of advanced cases. Methods: DNA was extracted from paraffin embedded tissues of advanced SGC and comprehensive genomic profiling (CGP) was carried out to evaluate for base substitutions, short insertions, deletions, copy number changes, gene fusions and rearrangements. Tumor mutation burden (TMB) was calculated on approximately 1.25 Mb. Some 324 genes in the FoundationOne CDX panel were analyzed. Results: Mucoepidermoid carcinoma (MECa) mutations were assessed. CDKN2A and CDKN2B GA were common in mucoepidermoid carcinoma (MECa) (52.5 and 30.5%). PIK3CA was also common in MECa (16.9%). ERBB2 amplification/short variants (amp/SV) were found in MECa (5.9/0%). HRAS GA was common in MECa (14.4%) as well. Other targets, including BAP1, PTEN, and KRAS, were noted but had a low incidence. In terms of immunotherapy (IO)-predictive markers, TMB > 10 was more common in MECa (16.9%). PDL1 high was also seen in MECa (4.20%). Conclusion: SGC are rare tumors with no FDA-approved treatment options. This large dataset reveals many opportunities for IO and targeted therapy contributing to the continuously increased precision in the selection of treatment for these patients.

## 1. Introduction

Malignant salivary gland carcinomas (SGC) are an uncommon group of head and neck cancers [1,2,3]. One of the most frequent types of SGC is mucoepidermoid carcinoma (MECa). Due to the rarity of the disease, data are often derived from case reports and retrospective series, rather than prospectively performed clinical trials. Thus, it has been challenging to define the role of chemotherapy or immunotherapy in management of metastatic or recurrent SGC.

Many possible biological targets have been identified. These include but are not limited to c-Kit [4], EGFR [5], HER2 [6,7], androgen, estrogen and progesterone receptor protein expression by immunohistochemistry [8], and PIK3CA [9] and BRAF mutations [7].

Although targeted therapies with kinase inhibitors [10,11,12] and monoclonal antibodies [13,14] have generally had low response rates, these therapies were given to unselected patients rather than matched to individuals whose tumors harbored equivalent aberrations [10,11,12,13,14]. On the contrary, when patients were appropriately treated for the presence of PIK3CA or ERBB2/HER2 alterations, significant responses have been described with alpelisib, trastuzumab and lapatinib [15], or mTOR inhibitors [16].

Since we are moving away from a one-size-fits-all approach to a tailored approach to SGC, further molecular understanding of salivary gland tumors is necessary. Here, we report the most frequent aberrations in two subtypes of SGC.

## 2. Methods Institutional Review Board (IRB)

Approval for this study was obtained from the Western IRB (Protocol No. 619478). A retrospective database search of a CLIA-certified and CAP-accredited reference molecular laboratory was performed for 1757 of SGC tissue samples. Clinicopathological data confirming that all cases were clinically advanced and metastatic SGC, including patient age and gender, routine histology and immunohistochemical staining results and confirmation of the diagnosis, were extracted from medical records and pathology reports. One (hematoxylin and eosin) H&E slide, corresponding to the tissue that was submitted for genomic profiling, was available for each case to review morphological features.

### 2.1. Cohort

Comprehensive genomic profiling of all 1757 SGC tissue samples was performed using a commercially available FDA-approved hybridization-captured, adaptor ligation–based libraries using DNA and RNA extracted from formalin-fixed paraffin-embedded tumor in a CLIA- and CAP-certified laboratory (FoundationOneCDx, Foundation Medicine, Inc., Cambridge, MA, USA). On central pathology review, all samples forwarded for DNA extraction contained a minimum of 20% tumor cells. The samples were assayed using adaptor-ligation and hybrid capture next-generation sequencing for all coding exons from up to 324 cancer related genes, plus select introns from up to 31 genes frequently rearranged in cancer. Patient samples were sequenced and evaluated for genomic alterations including base substitutions, insertions, deletions, copy number alterations (amplifications and homozygous deletions), and for select gene fusions/rearrangements, as previously described [17]. The bioinformatics processes used in this study included Bayesian algorithms to detect base substitutions, local assembly algorithms to detect short insertions and deletions, a comparison with process-matched normal control samples to detect gene copy number alterations and an analysis of chimeric read pairs to identify gene fusions, as previously described [17]. The tumor mutational burden was determined on 0.83–1.14 Mb of sequenced DNA using a mutation burden estimation algorithm that, based on the genomic alterations detected, extrapolates to the exome or the genome as a whole, as previously described [18]. In this study, low tumor mutational burden scores were defined as mean TMB, median TMB, percentage of cases with TMB ≥ 10 mutations/Mb, and percentage of cases with TMB ≥ 20 mutations/Mb. Assessment of microsatellite instability was performed from DNA sequencing across 114 loci, as previously described [19]. Each microsatellite locus had repeat length of 7–39 bp. The next-generation sequencing- based “microsatellite instability score” was translated into categorical microsatellite instability high, microsatellite instability intermediate, or microsatellite stable by unsupervised clustering of specimens, for which microsatellite instability status was previously assessed via gold standard methods. Only cases related to muco-epidermoid carcinoma will be discussed in this excerpt.

### 2.2. Laboratory Methods

PD-L1 expression was determined on subsets of the tumors using the DAKO 22C3 assay, with low positive tumor cell scoring defined as 1–49% staining and high positive tumor cell scoring defined as ≥50% staining.

### 2.3. Statistical Analysis

Fisher’s exact test was used in the statistical comparisons of the MECa with the other previously published groups of salivary gland carcinomas. Statistical significance was defined as *p* < 0.05.

## 3. Results

### Mucoepidermoid Carcinoma

Out of a total of 1757 cases, there were a total of 118 MECa tumors having 4.88 genomic alterations (GA)/tumor (Table 1). The median age of patients was 64, with a range of 16 to 89 years. Multiple mutations were found to be present in genes related to cell cycle regulation, chromosome, and chromatin remodeling along with different pathway-related genes. Concerning GA related to cell cycle regulation, 40.70% (*n =* 48) *TP53* GA, 52.50% (*n =* 62) *CDKN2A* GA, 30.50% (*n =* 36) *CDKN2B* GA, 3.40% (*n =* 4) were found in each of *CCND1* and *RB1* GA, while chromosome remodeling related genets such as *TERT* and *ARID1A* were found in 15% (*n =* 18) and 2.50% (*n =* 3), respectively. *BRCA1*, *BRCA2*, *ATM*, *PALB2*, and *BAP1* were the DNA damage response GA found in our tumor samples. Only 0.80% (*n =* 1) tumor was found to have *PALB2*, while 18.60% (*n =* 22) were found to have *BAP1*. *BRCA1*, BRCA2, ATM were found in 1.70% (*n =* 2), 5.90% (*n =* 7), 4.20% (*n =* 5), respectively. Nine tumors, 7.60%, were found to have *PTEN*, and 20 tumors, 16.90%, were found to have *PIK3CA* mutations. *NF1* and *TSC2* were found to be present in 5 and 1 tumors, respectively. *ERBB2* and *EGFR* were found in 5.9% (*n =* 7) and 0.8% (*n =* 1), respectively compared to *FGFR1* and KIT, which were found in 5.1% (*n =* 6) and 0.80% (*n =* 1), respectively. *NOTCH1* and *NOTCH2* were found in 4.20% (*n =* 5) and 7.60% (*n =* 9), respectively. *MTAP* and *MYC* were, respectively found in 16 tumors, 13.70%, and 2 tumors, 1.70%. GA related to RAS-RAF pathway were as follows: 3 tumors had KRAS G12C as a GA, while 17 had HRAS GA and 2 had BRAF GA.

A total of 100 cases of MECa tumors had their immuno-oncology biomarkers investigated. None had a reported MSI high frequency. Similarly, CD274 (PD-L1) Amp was not detected in any tumors as well. The median TMB observed was 2.6; TMB > 10% was 16.90% and TMB > 20% was 12.70%. PD-L1 low expression was seen in 26 cases vs. high expression, which was 4.20%. MDM2 Amp and STK11 inactivating GA were each observed in 3.4%, *n =* 3 tumors.

## 4. Discussion

Mucoepidermoid carcinoma (MECa), the most common malignant salivary gland neoplasms across all ages, are histologically composed of squamoid, mucin-producing and intermediate cell types, which may form various patterns, including cystic and/or solid areas [20,21]. Some of the variants which have been identified include oncocytic, clear cell, sebaceous, and sclerosing variants [21]. According to the World Health Organization (WHO), MECa are divided into three groups: low, intermediate, and high histologic grade, with different clinical courses [21]. Several systems suggest stratifying MECa according to prognostic significance. Some of these include the Brandwein system, the Armed Forces Institute of Pathology scoring system, the Katabi stratification method, and the modified Hailey system [22,23].

In a study performed by Kato et al., 117 patients with salivary gland tumors were assessed, only five of which had MECa tumors [24]. Out of these patients, GA in each of *TP53, PI3K*, *BAP1*, and *PTEN* were found in two patients. GA in PIK3CA, HRAS, NF1 and the cyclin-dependent pathway, which included the following aberrations: *CCNDI, CDK4/6 or CDKN2A/B,* were found in one patient. Compared to this, our study had a higher percentage of the aforementioned mutations.

### 4.1. Genetic Alterations

TP53 is a marker of aggressive behavior and poor overall response in all malignancies in general [25]. TP53 alterations were historically reported in 25–33% of MEC, although the sample size was limited in these studies [26]. Compared to this, our study reported TP53 alteration in 40.7% of the patients. TP53 alterations are preferentially found in high grade tumors, indicating that they may represent a switch for transformation from low to high grade [27].

Some reports have indicated that EGFR overexpression may be found in 73% of high grade MEC and connotes a poor prognosis. However, EGFR alterations are generally rare in salivary gland tumors overall and were barely noted in our dataset [27]. Drugs such as Sotorosib have promising anti-cancer activity in KRAS G12C mutated advanced solid cancers [28]. KRAS G12C represents around 14.5% of all KRAS alterations in all tumors [29]. This was found in 2.5% of our cohort of MEC. PIK3CA with agents such as Alpelisib [R6] was found in 16.8% in our cohort. AR alteration, which is frequent in other salivary gland tumors such as salivary ductal carcinoma, was not found in MEC in our analysis [30].

The most significant genetic alteration in MECa is characterized by t(11;19)(q21;p13) translocation which results in the fusion of CREB regulated transcription coactivator 1(*CRTC1*)*-* mammalian mastermind-like gene (*MAML2*) [31]. This translocation, which acts as a potential main driver for mutations, can be detected in 55–88% of MECa [32,33,34]. CAMP response element-binding protein (CREB) is encoded by the *CRTC1* gene which is located on chromosome 19. *CRTC1*, which is also known as *MECT1*, *TORC1*, and *WAMTP1*, modulates cellular pathways mediated by activating protein 1 (AP-1) [32,34,35,36]. *CRTC1*’s binding to CREB enhances its transcription, thus allowing it to carry out its role in gene cellular proliferation and differentiation. Here, it is worth mentioning that when a fusion occurs in exon 1 of the *CRTC1* gene and exon 2 to 5 of the *MAML2* gene, the CREB-binding domain of *CRTC1* replaces the notch-binding domain of *MAML2,* thus activating the notch pathway [33,34,35,36]. This newly formed fusion protein activates transcription of cAMP target genes, including *PEPCK1*, *AREG*, *MMP10*, *IL6*, *NR4A2*, and *NR4A3*. AREG detection using immunohistochemistry may help identify fusion-positive MECa [37]. MECa tumorigenesis is thought to arise from the interaction of the fusion gene with *MYC* and *AP-1* [38,39]. Another translocation in rare types of MECa occurs between the *EWSR1* and *POU5F1* genes [40].

*CRTC3-MAML2* gene fusion is less commonly detected in a small subset (around 5%) of MECa [39]. Studies have shown that *CRTC1-MAML2* and *CRTC3-MAML2* mutations are mutually exclusive [36]. An unfavorable prognosis has been reported in *CRTC1-MAML2* fusion-positive MECa with *CDKN2A* deletions [41].

### 4.2. Diagnostic Markers

MECa diagnosis can be carried out solely using histology, especially when the tumor shows the typical admixture of squamoid, mucinous, and intermediate cells. When histology is not sufficient for diagnosis, genetic or molecular tests can be utilized. Genetic tests for rearrangements can be helpful to distinguish a Warthin tumor from the Warthin-like variant of MECa and other clear cell maligancies of the salivary gland [42]. Molecular tests, such as a reverse transcription polymerase chain reaction or fluorescent in situ hybridization (FISH) aid in diagnosing variants of MECa, such as oncocytic or clear cell types [43,44,45]. A translocation can be detected by these techniques, especially when FISH and break-apart probes are designed to detect genes. Furthermore, using these techniques, the *MAML2* rearrangement can be detected. The *CRTC1/3-MAML2* fusions are the only accepted diagnostic markers for MECa. This trans-location may also be detected in cutaneous hidroadenomas that have histologic similarities to MECa [46].

### 4.3. Prognostic Markers

Earlier, several studies have revealed that a favorable prognosis is associated with *CRTC1/3-MAML2* gene rearrangements in younger people, and with a tumor of low to intermediate grade [38,47,48,49,50]. Recently, studies have shown that these mutations are not independent prognostic markers and are not related to prognosis or tumor grade. This can be explained by several reasons. First, some tumors are not diagnosed as MECa due to strict diagnostic criteria. Second, the prevalence of the translocation in MECa varies based on stringency of histologic diagnosis. Last, previous studies likely did not account for the confounding effect of grade on survival [51,52,53,54].

Purkinje cell protein (PCP) 4/peptide (PEP) 19 is a calmodulin-binding antiapoptotic peptide. Yoshimura et al. found that PCP4/PEP19 expression was related to a better prognosis, while HER2 expression was associated with a worse prognosis [55].

### 4.4. Treatment

Treatment of advanced salivary gland tumors remains limited to systemic chemotherapy, especially in the absence of molecular targets. Given the low response and survival rates with conventional systems, it becomes even more important to identify potential targets. Although our study does not provide any information on treatment outcomes, identifying and describing potential GAs can be of help in the future to identify potential targeted agents [56].

The mainstay of treatment of MECa, similar to any other salivary gland tumor, is surgical resection with disease-free margins [57,58,59]. Better regional control was achieved with combined chemoradiotherapy, but no survival difference was found compared to patients receiving radiotherapy alone [60]. As a matter of fact, systemic therapies should be used as palliative treatment for cancer-related symptom relief, or in rapid progression of the disease [61,62,63].

#### Standard Therapy

i.Surgery

The most critical factor in determining the prolonged outcome and disease-specific survival is locoregional disease control [64]. To enhance locoregional control, postoperative radiotherapy is recommended, and it is typically reserved for cancers with high-risk characteristics. High-risk characteristics are defined by close or positive surgical margins, nodal metastases, extracapsular spread (ECS), perineural invasion, lymph vascular invasion, advanced tumor (T) stage, and high-grade histopathology [65,66]. In cases of deep lobe and recurrent cancers, radiation therapy may also be utilized. Doses greater than 60 Gy are necessary to achieve local tumor control. Higher doses, along with chemotherapy or alone, may be used to treat medical or technically unresectable tumors [67].

To our knowledge, there have only been two retrospective studies solely observing MECa, one describing the epidemiology and the other dealing with prognostic factors [68,69]. Most salivary gland tumor studies have been on a small series of patients with different salivary gland tumors treated uniformly with single agent chemotherapy or a combination of agents.

ii.Chemotherapy

Airoldi et al. evaluated cisplatin plus vinorelbine (VNB) in 16 patients with recurrent metastatic salivary gland tumors; only one had MECa, vs. VNB alone in 20 patients. The combination of cisplatin plus VNB was found to be superior to VNB, but a prognosis related to the sole MECa tumor could not be singled out [70]. Gilbert et al. conducted a phase II evaluation of single-agent paclitaxel 200 mg/m^2^ every 3 weeks, which resulted in eight partial responses among 31 patients with mucoepidermoid or adenocarcinoma histologic subtypes, giving a response rate of 26% with a 95% confidence interval of 10% to 41% [71]. The last chemotherapeutic regimen which was also found to be useful consisted of docetaxel. Raguse et al. determined the role of docetaxel in four patients with high-grade mucoepidermoid cancer of the major salivary glands. After six cycles, complete remission was noticed in two patients, and partial remission was seen in the other two patients [72].

iii.Monoclonal antibody

Many studies suggest amplification/overexpression of HER2/neu in mucoepidermoid carcinomas [73,74,75,76]. HER2 positivity found by IHC has been reported in 5.5% of MECs [77]. In our study, HER2 amplification was found in 5/9% of patients. Haddad et al. assessed the use of Herceptin (trastuzumab) on 14 patients with salivary gland tumors (SGC), three of which had MECa tumors with overexpressed HER2/neu in their salivary gland tumors. A partial response was seen only in one out of three patients, and lasted more than 2 years [14].

A case report by Gazola et al. revealed how a patient who was refractory to trastuzumab, ado-trastuzumab emtansine and tucatinib actually achieved excellent clinical benefit from Fam-trastuzumab deruxtecan [78]. Tsurutani et al. assessed the safety and tolerability of trastuzumab deruxtecan in pretreated, Her2-expressing non-breast/non-gastric or Her2-mutatnt solid tumors. Out of 60 patients, 8 patients had salivary gland tumors. Unfortunately, the results were not broken down into distinct tumor types. However, it was reported that tumor types including HER2-expressing or *HER2*-amplified salivary gland cancer, biliary tract cancer, and endometrial cancer, and *HER2*-mutant nonamplified breast cancer had a median progression-free survival of 11.0 months (95% CI, 2.8-Not evaluated) [79]. Currently, there are 8 clinical trials assessing trastuzumab deruxtecan or trastuzumab emtansine in salivary gland tumors.

The National Cancer Institute—Molecular Analysis for Therapy Choice (NCI-MATCH) assessed the efficacy of ado-trastuzumab emtansine (T-DM1) in HER2-amplified histologies in MECa. A partial response was observed in a patient with MECa [80]. Similarly, a case report of an elderly male with HER-2 Neu-overexpressing metastatic MECa demonstrated a prompt and sustained disease response to targeted therapies directed against HER-2 Neu, with a long survival interrupted by hepatoxicity to TDM-1 treatment [81].

iv.Targeted therapies

1.Sorafenib

It is known that MECa has high angiogenic activity with increased expression of vascular endothelial growth factor (VEGF) and angiopietin-2 (ANG2) [82,83,84]. In a phase II trial with recurrent and malignant salivary gland tumors, Locati et al. reported that sorafenib demonstrated a rapid decrease in disease progression in two patients with MECa [85].

2.Nintedanib

In a single-arm, phase II trial carried out by Kim et al., 20 patients with SGT were enrolled, of which two patients (10%) had MECa tumors [86]. Results were not segregated by histological subtype, but 75% of patients had stable disease.

3.Lapatinib

According to Lujan et al., the EGFR pathway is activated in high-grade MECs with aggressive behavior [87]. However, our report revealed very small numbers. Inhibiting both EGFR and receptor ERBB2 receptors, lapatinib was found to have cytostatic effect. In a phase II study performed by Agulnik et al., no objective response was noticed in the two MECa patients; however, stable disease (>6 months) was observed in 36% of all the eligible patients, *n =* 62 [12].

4.Vorinostat

Pouloudi et al. were the first to assess immunohistochemically the expression of Hdi in SGC [88]. Wagner et al. have recently assessed acetyl-histone H3 expression (AH3) and Ki67 index in tissue microarrays (TMAs) of 84 cases of SGC [89]. Ahn et al. assessed the inhibition of histone deacetylase (HD)-7 expression in apicidin-treated MEC cell lines, providing evidence that HD-7 downregulation inhibits cell proliferation and induces autophagy in MEC cells [90]. Wagner et al. also investigated the benefits of Hdi and NFκB combined inhibition in MECa cell lines with the administration of Vorinostat and Emetine, respectively [89]. Vorinostat failed in significantly reducing the total number of tumor cells. Alone, Vorinostat efficiently disrupted the population of cancer stem cells. However, when combined with Emetine, Vorinostat provided an effective regimen for managing MECa [89].

5.ANA-12

ANA-12 is a tyrosine receptor kinase B (TrkB) inhibitor. The brain-derived neutropenic factor (BDNF) is a growth factor that binds to TrkB and activates downstream pathways such asPI3K/Akt, which has a crucial role in tumorigenesis [91]. Perineural invasion in high-grade MECa, another negative prognostic marker, is associated with expression of BDNF and TrkB [92,93]. In vitro studies have shown that inhibiting TrkB decreases invasion and delays migration of MECa cells. Combining ANA-12 with cisplatin, however, caused the recovery and re-accumulation of cancer stem cells [92,93].

v.Immunotherapy

TMB is a measure of neoantigen burden and predicts responsiveness to immune checkpoint therapy [94]. Similarly, PD-L1 expression estimated by immunohistochemistry assays can also predict immunotherapy responsiveness [95]. The multi-cohort phase 2 KEYNOTE-158 study established the role of pembrolizumab in patients with TMB > 10, resulting in the FDA issuing a blanket approval in this setting, regardless of the tumor type [96]. Upon analysis of a cohort of 109 advanced salivary gland carcinomas from the KEYNOTE-158 trial, pembrolizumab had an objective response rate of 4.6% in the overall population and 10.7% in the PD-L1 positive (CPS ≥ 1) cohort. Overall survival was 21 months and progression-free survival was 4 months [97]. KEYNOTE-028 was a phase 1 trial of 26 advanced salivary gland tumors with PD-L1 ≥ 1%. The objective response rate was 12%, and median duration of response was 4 months [98]. While trials specifically studying the role of checkpoint inhibitors in MECa are lacking [64], individual case reports have demonstrated durable clinical responses with checkpoint therapy [99]. Our report highlights that in some patients with advanced or refractory MECa having TMB > 10 or PD-L1 positivity, immune checkpoint therapy may be a viable option.

vi.Novel studies

*CRTC1-MAML2* positive cells were sensitive to epidermal growth factor receptor (EGFR) tyrosine kinase inhibition pre-clinically, suggesting a potential role for such drugs [100]. *ERBB2* amplifications, though infrequent, may be amenable to Her-2 targeted therapy [101].

Last, Wein et al. reported that the combination of the notch inhibitor, dnMAML1 and the γ-secretase inhibitor, and the EGFR inhibitor Erlotinib enhanced MECa suppression as compared to individual treatment [102].

## 5. Synopsis

Treatment requires a multidisciplinary approach, and surgery to negative margins is a mainstay, supplemented by radiation for better locoregional control and chemotherapy, usually in the palliative recurrent or metastatic setting. Currently, surgical management and adjuvant radiotherapy are the best options for achieving disease control because conventional chemotherapies are ineffective against this disease due to resistance [103,104,105,106].

The high recurrence rate seen in most studies suggests that initial surgery should be radical with sufficient normal tissue margins [107]. Preservation of the facial nerve should be suggested only when the parotid tumor is small and is situated far away from the nerve. Otherwise, in cases of extensive tumor invasion, total parotidectomy with partial resection of the facial nerve is preferred [107].

Lately, there has been a huge emphasis on targeted therapy. Ross et al. conducted a comprehensive analysis of genomic profiles of metastatic and relapsed salivary gland carcinomas [108]. The MECa in the current study featured GA that were most similar to ductal adenocarcinomas and adenocarcinomas not otherwise specified including similar frequencies of alterations in ERBB2, PIK3CA, BRAF and TP53 (all differences not significant) [108].

In the literature, some cases of response to immune checkpoint inhibitors exist. Despite having no PD-L1 expression, two cases of metastatic high-grade MECa with prolonged response to immune checkpoint inhibitor pembrolizumab were reported by Pharaon et al. [99]. This shows the necessity of conducing further genomic profiling to unravel the connection between immunotherapy and the tumor microenvironment.

## 6. Conclusions

As rare and highly malignant tumors, SGC have a few proven target therapeutics. As mentioned above, several predictive molecular markers are being unraveled. Under the current treatment guidelines, most unresectable tumors become resistant to treatment within a short period of time. Our data suggest that MECa is a heterogeneous disease at the genomic level. Closer examination of the genomic profile also suggests that many of the MECa samples possess at least one target gene with direct or related targeted inhibitors currently available. Thus, it is important to translate knowledge obtained from genomic analysis of SGC samples into clinical cancer sequencing, combined with precision oncology clinical trials.

## Figures and Tables

**Table 1 biomedicines-11-00519-t001:** Clinical characteristics and genomic alterations in MECa.

**Muco-Epidermoid Carcinoma**	
Number of cases	118
Median age (range)	64 (16–89+)
Gender (M/F)	58%/42%
GA per tumor	4.88
**Cell Cycle Regulatory GA**	
*TP53*	40.70%
*CDKN2A*	52.50%
*CDKN2B*	30.50%
*CDK4*	1.70%
*CCND1*	3.40%
*RB1*	3.40%
**Chromosomal and Chromatin Related GA**	
*TERT*	15.00%
*ARID1A*	2.50%
**xRAS-RAF Pathway GA**	
*KRAS* All	5.10%
*KRAS* G12C	2.50%
*HRAS*	14.40%
*BRAF*	1.70%
**MTOR Pathway GA**	
*PTEN*	7.60%
*PIK3CA*	16.90%
*NF1*	4.20%
*TSC2*	1.00%
**DNA Damage Response Associated GA**	
*BRCA1*	1.70%
*BRCA2*	5.90%
*ATM*	4.20%
*PALB2*	0.80%
*BAP1*	18.60%
**Receptor Tyrosine Kinase Targetable GA**	
*ERBB2* (amp/SV)	5.9%/0%
*EGFR* (amp/SV)	0.8%/0%
*FGFR1*	5.10%
*FGFR2*	0%
*RET*	0%
*ETV6lNTRK3* fusion	0%
*MET*	0%
*KIT*	0.80%
**Transcription Factor Genomic Alterations**	
*NFIB-MYB* Fusion	0%
*ESR1*	0%
*AR*	0%
*MYC*	1.70%
*EWSR*	0%
**Emerging Potentially Genomic Alterations**	
*NOTCH1*	4.20%
*NOTCH2*	7.60%
*MTAP*	13.70%
(F1CDx only)	(51 cases)
**Immuno-Oncology Drug Biomarkers**	
MSI high frequency	0%
	(100 cases)
*CD274* (*PD-L1*) amp	0%
*STK11* inactivating GA	3.40%
*MDM2* amp	3.40%
Median TMB	2.6
TMB > 10%	16.90%
TMB > 20%	12.70%
PD-L1 low expression (≤49%)	38.40%
	(26 cases)
PD-L1 high expression (>50%)	4.20%

## Data Availability

The original contributions presented in the study are included in the article, further inquiries can be directed to the corresponding author/s.

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
