# Peer review of "Molecular Targets in Salivary Gland Cancers: A Comprehensive Genomic Analysis of 118 Mucoepidermoid Carcinoma Tumors"

_biomedicines, 2023, doi:10.3390/biomedicines11020519_

Round 1

Reviewer 1 Report

Known in the field based on previous literatures:

1.  Salivary gland carcinoma (SGC) is a rare tumor of head and neck cancers. Neoplastic tumors are often heterogeneous and contain more than one type of cell, including mucoepidermoid carcinoma (MECa) and adenoid cystic carcinoma (AdCC). MECa is the most common malignant salivary gland tumor, followed by AdCC.

2. The prime cause of disease comprises changes in the DNA sequence (mutation), copy number variations, and gain or loss of different genes.

In this article authors reported following findings:

I have gone through the article titled “Molecular targets in salivary gland cancers: A comprehensive genomic analysis of 118 Mucoepidermoid Carcinoma Tumors”. Authors extracted the DNA from tissues of 16 advanced SGC and comprehensive genomic profiling (CGP) was performed to evaluate the changes in DNA sequence (base substitutions, short insertions, deletions, copy number changes, gene fusions and rearrangements). Authors are performed and analyzed following findings-

1. Authors performed the clinical characterization and genomic alterations in mucoepidermoid carcinoma.

2. Authors discussed the different mode of therapies including chemotherapy, monoclonal antibody, and targeted therapies.

The data presented are interesting and generally supportive of the conclusions drawn. There are, however, several issues that require the authors' attention. The following suggestions if incorporated could help in the better understanding of the significance of the work and implications.

 Minor/major Concerns:

1.     The methods part is not sufficient and clear hence need to rewrite. What is IRB in line 53? Authors should write meaning.

2.     In methods, line 60- what is H&E? Authors should write meaning of abbreviation.

3.     What is meaning of GA in GA/tumor (line 102)?

4.     Lines 196-199 are not clear. Please reframe it.

5.     In line 199-200, the sentence is not clear. Authors should add ‘by” to complete the lines. So, lines should be- A translocation can be detected by these techniques, especially when FISH and break-apart probes designed to detect genes.

6.     Have you performed the GO analysis of differential expressed genes associated with MECa to know the GO function and pathways? How much similarity and dissimilarity in number of genes, sign, and symptoms you have observed in the current study as compared to previous reported case?

7. Therapy part in discussion is well discussed but it’s so much elaborated, hence, need to be consign.

8. There are many studies for MECa and AdCC are available. Explain, how your study is different from rest excluding identification and matching of some genes? Does it address a specific gap in the field?

Author Response

Minor/major Concerns: 

  1. The methods part is not sufficient and clear hence need to rewrite. What is IRB in line 53? Authors should write meaning.

 IRB has been mentioned. If the reviewers can point out which part needs to be rewritten, please let us know. The methods section has been re addressed. We believe it portrays all the steps made to conduct the research done in this manuscript.

  1. In methods, line 60- what is H&E? Authors should write meaning of abbreviation.
    1. Done
  2. What is meaning of GA in GA/tumor (line 102)?
    1. Done
  3. Lines 196-199 are not clear. Please reframe it.
    1. Done
  4. In line 199-200, the sentence is not clear. Authors should add ‘by” to complete the lines. So, lines should be- A translocation can be detected bythese techniques, especially when FISH and break-apart probes designed to detect genes.
    1. Done
  5. Have you performed the GO analysis of differential expressed genes associated with MECa to know the GO function and pathways? How much similarity and dissimilarity in number of genes, sign, and symptoms you have observed in the current study as compared to previous reported case?

Yes. MECa featured genomic alterations that were most similar to ductal adenocarcinomas and adenocarcinomas not otherwise specified including similar frequencies of alterations in ERBB2, PIK3CA, BRAF and TP53 (all differences not significant) (108).

  1. Therapy part in discussion is well discussed but it’s so much elaborated, hence, need to be consign.

Sections on therapy and treatment has been shortened

  1. There are many studies for MECa and AdCC are available. Explain, how your study is different from rest excluding identification and matching of some genes? Does it address a specific gap in the field?

A paragraph comparing our results to other studies have been added. Since there is little information on certain genetic alterations in MEC, our study adds to the available literature, and may help future studies identify potential therapeutic targets.

if granted more time there are a few edits which we may want to do.

Reviewer 2 Report

The authors analyzed gene profiling of salivary mucoepidermoid carcinomas (MECa) using NGS. The results of 118 MECas are very important, helping head and neck oncologists to assess new strategies for MECa. Some issues below should be reconsidered.

As the authors mentioned, MECa is classified into low-, intermediate-, and high-grade histologically. The results including GA percentage should be described separately for each three group like reference 37.   

The Discussion part includes comprehensive literatures, digesting lots of information like a review. It is really valuable for the readers. But the authors should assess more their own results referring the previous papers here.

Line 21 GA should be explained (Its first showing up).

Line 51, "two histological subtypes". What does it mean?

Line 82-84, "low tumor mutational burden scores were defined as mean TMB, median TMB, percentage of cases with TMB > 10 mutations/Mb and percentage of cases with TMB > 20 mutations/Mb."  I can not understand meaning of these sentences.

Line 97, comparisons of 2 groups? Where are results of statistical analyses?

Line 106, 1.70% is shown as  30.50% in Table 1.

Line 140-142, "While all patients were found to have GA in the PI3K pathway, only 2 were found to have GA in TP53, PIK3CA, or HRAS. One patient only had a GA in PTEN." This is misunderstanding the results of ref 24.

Line 182, again is a gain?

Line 188, adenocarcinoma should be MECa.

Line 196,   (38). ?

In Discussion, the paragraph numbers or characters are strange. They should be arranged with regularity.

Line 259, salivary gland tumors (SGT) should be unified to SGC.

Line 279, In our... ?

Line 288, 8intedanib.

Line 298, receptor ERBBB2 receptors, it seems to be strange.

Line 299 and 315, MEC should be MECa.

Line 315, HD-7 needs explanation (its first showing up).

Line 342, duration was 4% ?

Line 365, "Otherwise in cases of extensive tumor invasion".  This sentence should be sophisticated.

Line 369, MECs should be revised.

Line 391,  ( ) ?

Author Response

The authors analyzed gene profiling of salivary mucoepidermoid carcinomas (MECa) using NGS. The results of 118 MECas are very important, helping head and neck oncologists to assess new strategies for MECa. Some issues below should be reconsidered.

  • As the authors mentioned, MECa is classified into low-, intermediate-, and high-grade histologically. The results including GA percentage should be described separately for each three group like reference 37.  
  • In this study histologic grading of the MECa was not performed.  
  • The Discussion part includes comprehensive literatures, digesting lots of information like a review. It is really valuable for the readers. But the authors should assess more their own results referring the previous papers here.

information on how some of our results compares with the available literature has been added

  • 21 GA should be explained (Its first showing up).
    • Done

  • Line 51, "two histological subtypes". What does it mean?
    • It has been fixed to two subtypes

  • Line 82-84, "low tumor mutational burden scores were defined as mean TMB, median TMB, percentage of cases with TMB > 10 mutations/Mb and percentage of cases with TMB > 20 mutations/Mb."  I can not understand meaning of these sentences.

This is a common method of defining TMB. The mean TMB on occasion can be distored when cases of extremely high TMB (i.e. > 100 mutations/Mb) are included in this calculation. For this reason, the median TMB is also provided. The TMB cut-off of > 10 mutations/Mb is included as it is the FDA approval cut-off for pembrolizumab in a pan-cancer setting that would include salivary gland carcinomas.

  • Line 97, comparisons of 2 groups? Where are results of statistical analyses?

The following sentence has been added to the discussion (lines 375-378).

“MECa in the current study featured genomic alterations that were most similar to ductal adenocarcinomas and adenocarcinomas not otherwise specified including similar frequencies of alterations in ERBB2, PIK3CA, BRAF and TP53 (all differences not significant) (108).

  • Line 106, 1.70% is shown as  30.50% in Table 1.
    • Done
  • Line 140-142, "While all patients were found to have GA in thePI3K pathway, only 2 were found to have GA in TP53, PIK3CA, or HRAS. One patient only had a GA in PTEN." This is misunderstanding the results of ref 24.
    • Done
  • Line 182, again is a gain?
    • Done
  • Line 188, adenocarcinoma should be MECa.
    • done
  • Line 196,   (38). ?
    • Done
  • In Discussion, the paragraph numbers or characters are strange. They should be arranged with regularity.
    • Done
  • Line 259, salivary gland tumors (SGT) should be unified to SGC.
    • Done
  • Line 279, In our... ?
    • Done
  • Line 288, 8intedanib.
    • Done
  • Line 298, receptor ERBBB2 receptors, it seems to be strange.
    • Done
  • Line 299 and 315, MEC should be MECa.
    • Done
  • Line 315, HD-7 needs explanation (its first showing up).
    • Done
  • Line 342, duration was 4% ?
    • done
  • Line 365, "Otherwise in cases of extensive tumor invasion".  This sentence should be sophisticated.
    • Done
  • Line 369, MECs should be revised.
    • Done
  • Line 391,  ( ) ?
    • done

IF we have more time, there are a few edits we would like to do.